# RMD and Its Suppressor MAPK6 Control Root Circumnutation and Obstacle Avoidance via BR Signaling

**DOI:** 10.3390/ijms251910543

**Published:** 2024-09-30

**Authors:** Le Dong, Jianxin Shi, Staffan Persson, Guoqiang Huang, Dabing Zhang

**Affiliations:** 1Joint International Research Laboratory of Metabolic & Developmental Sciences, State Key Laboratory of Hybrid Rice, School of Life Sciences and Biotechnology, Shanghai Jiao Tong University, Shanghai 200240, China; jianxin.shi@sjtu.edu.cn (J.S.); staffan.persson@plen.ku.dk (S.P.); huang19880901@sjtu.edu.cn (G.H.); zhangdb@sjtu.edu.cn (D.Z.); 2Department of Plant & Environmental Sciences, Copenhagen Plant Science Center, University of Copenhagen, 1871 Frederiksberg, Denmark

**Keywords:** wavy root, root circumnutation, obstacle avoidance, RMD, MAPK, brassinosteroid, rice

## Abstract

Helical growth of the root tip (circumnutation) that permits surface exploration facilitates root penetration into soil. Here, we reveal that rice actin-binding protein RMD aids in root circumnutation, manifested by wavy roots as well as compromised ability to efficiently explore and avoid obstacles in *rmd* mutants. We demonstrate that root circumnutation defects in *rmd* depend on brassinosteroid (BR) signaling, which is elevated in mutant roots. Suppressing BR signaling via pharmacological (BR inhibitor) or genetic (knockout of BR biosynthetic or signaling components) manipulation rescues root defects in *rmd*. We further reveal that mutations in *MAPK6* suppress BR signaling and restore normal root circumnutation in *rmd*, which may be mediated by the interaction between MAPK6, MAPKK4 and BR signaling factor BIM2. Our study thus demonstrates that RMD and MAPK6 control root circumnutation by modulating BR signaling to facilitate early root growth.

## 1. Introduction

Successful establishment of seedlings requires primary roots to penetrate the soil efficiently, thereby overcoming challenges such as varied soil substrates or complicated environmental conditions. Root circumnutation (helical movement) was first observed by Charles and Francis Darwin, and has been proposed to play an important role in obstacle avoidance [1]. Nevertheless, functional characterization of specific genes associated with root circumnutation has been limited until the recent discovery of a *rice histidine kinase-1* (*oshk1*) mutant, which confirmed that circumnutation facilitates root growth around obstacles and penetration of substrate [2]. OsHK1 is a positive regulator of ethylene signaling that acts downstream of ethylene receptors to inhibit root elongation [3]. OsHK1 supports root exploration of solid surfaces and obstacle avoidance by integrating cytokinin–auxin signaling pathways, directing differential cell elongation during circumnutation [2]. Although the identification of HK1 has provided a significant step towards understanding the molecular basis of root circumnutation in plants, the regulation of this growth process remains largely unknown.

The actin binding protein RMD belongs to the type II formin FH5, and is involved in nucleating actin assembly and bundling actin filaments [4,5]. RMD wraps the actin filaments around statoliths to buffer their movement during gravity sensing, and adjusts root angles in response to external phosphate availability [6]. In shoots, RMD maintains actin configurations that facilitate statolith mobility and promote negative gravitropism in response to light [7]. *rmd* mutants display pleiotropic phenotypes, including dwarfism, compromised cell expansion, shoot bending of uppermost internodes, curled seeds, and wavy root pattern [4,5]. Although it has been suggested that non-linear root growth may aid in obstacle avoidance in the soil [8,9], the precise mechanisms underlying this phenomenon are still poorly understood.

The plant hormone brassinosteroid (BR) affects many processes in root development, such as meristem size [10,11,12], initiation of lateral root primordia [13], and root hair formation [14]. An imbalance in BR levels is harmful to primary root growth and development, as evidenced by root elongation by low BR concentrations and suppression of root length by high concentrations [15]. BR treatment enhances the gravitropic curvature in the maize primary root [16]. Notably, BR treatment induces wavy root growth in *Arabidopsis* and rice [17,18,19]. This phenotype is similar to the wavy phenotype observed in Arabidopsis roots grown on slanted impenetrable medium, which was hypothesized to be a consequence of touch, gravity, and circumnutation, or as a result of the physical interaction between the root tip and the growth media, where medium friction impedes the root tip’s movement [20]. However, the actual underlying mechanism remains elusive. BR-induced wavy roots display cytoskeleton changes resembling those observed in Arabidopsis *act2-5* mutants, which also have elevated BR signaling, suggesting an interplay between BR and the cytoskeleton in root morphology [21]. While phytohormones such as ethylene, cytokinin, and auxin have been implicated in root circumnutation [2], the connection between BRs and circumnutation has not yet been elucidated. To elucidate the molecular basis underlying the wavy root phenotype observed in the *rmd* mutant and advance our understanding of root development regulation, this study conducted a suppressor screen for the wavy root phenotype in the *rmd* mutant. This approach successfully identified a suppressor of *rmd*, designated *sor8*, which corresponds to *MAPK6.* Based on this discovery, a novel BR-mediated regulatory mechanism for root circumnutation has been unveiled.

## 2. Results

### 2.1. RMD Controls Root Circumnutation

*rmd* mutants display a wavy root phenotype (Figure 1A); however, the underlying mechanism is unclear. Furthermore, we found that *rmd* roots exhibit a coiling behavior when reaching the bottom of a growth container, in contrast to the relatively straight growth pattern observed in WT roots along the surface (Appendix A). This coiling phenotype resembles that of the *hk1* mutant, where root tip circumnutation was disrupted [2]. To study root circumnutation of *rmd* roots, we conducted imaging of primary root growth in a gel-based (transparent) medium at 10 min intervals over a period of 3 h. WT primary roots circumnutate in a helical pattern before growing straight, repeating the process of bending and straightening (Figure 1B and Appendix A). By contrast, the *rmd* mutant roots exhibit normal bending, but do not straighten, maintaining wavy root architecture (Figure 1B and Appendix A). Analysis of cell wall staining in the roots revealed that cells on the inner side of curved roots in *rmd* mutants are significantly shorter than those on the outer edge, whereas cells in WT roots are of equal length on both sides along the roots’ length (Figure 1C,D). Isaiah et al. suggested that root tip circumnutation is driven by transient, circumferentially localized differential elongation between the inner and outer sides of the bending root [2]. In conjunction with PI staining, we propose that after the initial bending event, the persistent discrepancy in growth rates between the inner and outer sides of the root in *rmd* mutant maintains the helical trajectory of circumnutation along the length of the root (Appendix A).

### 2.2. Mutations in SOR8/MAPK6 Rescue Root Growth Defects of rmd Mutant

To further elucidate the function of RMD in root circumnutation, we performed a suppressor screen of the wavy root phenotype in an ethyl methanesulfonate (EMS)-mutagenized *rmd* population, resulting in the identification of *suppressor of rmd 8* (*sor8*) (Figure 1A). Cell wall staining of roots revealed that there was no significant difference in the cell length on the inner and outer sides of *rmd sor8* mutant roots (Figure 1C,D), indicating that the *sor8* mutation impedes differential cell growth. The circumnutation pattern of both *rmd sor8* and *sor8* roots was similar to WT roots (Figure 1B and Appendix A), implying a collaborative role of RMD and SOR8 in root circumnutation.

The *sor8* mutation was genetically identified through bulked-segregant analysis (BSA) of *rmd*-like and *rmd sor8*-like populations in the F_2_ progeny resulting from a cross between *rmd* and *rmd sor8*. Sequencing data predicted the genomic regions linked to the straight-root trait, with one of the most significant linkage positions at a missense mutation in *LOC_Os06g06090* (encoding the MAP kinase protein MAPK6), leading to a H349R substitution at the C-terminal end of the serine–threonine kinase domain of this protein (Figure 2A). Multiple sequence alignment revealed that the His^349^ residue in MAPK6 is highly conserved across species, suggesting that it may play a critical role in MAPK6 function (Figure 2B). The complementation of *rmd sor8* plants with a genomic *MAPK6* clone displayed the wavy root phenotype resembling that of *rmd* plants (Figure 2C), confirming the accurate identification of the causative *sor8* mutation.

### 2.3. Reduced BR Sensitivity by the mapk6 Mutation Alleviates rmd Phenotype

To investigate the regulatory role of MAPK6 in root growth, we undertook an immunoprecipitation–mass spectrometry (IP-MS) experiment to identify potential interacting proteins of MAPK6. The MAPK6 protein was precipitated from the *pMAPK6*:*MAPK6-GFP* complementation line in *sor8* plants, with parallel extractions from WT plants serving as a control to minimize false positives. MAPK6 peptides were highly enriched in the MAPK6-GFP sample (Figure 3A), and not detected in the WT control. The presence of MKK4 in the IP-MS results (Figure 3A) was anticipated, due to the well-documented role of the MKK4–MAPK6 module in various aspects of plant growth and development. For instance, the YDA (MAPKKK4)-MKK4/5-MAPK3/6 cascade regulates various developmental processes in *Arabidopsis*, such as stomatal development, cytokinesis, inflorescence architecture, abscission, root apical meristem development, and embryonic patterning [22]. Similarly, in rice, the OsMKKK10–OsMKK4–OsMAPK6 cascade controls the balance between grain size and grain number [22]. In this study, we found a potential interaction between MKK4 and MAPK6 in rice roots, suggesting a possible role in root morphogenesis.

We also identified the BR signaling proteins BLE (Brassinolide (BL)-enhanced gene) and BIM2 (BES1-interacting Myc-like2) in the MAPK6 interactome (Figure 3A). BIM2 is a basic helix–loop–helix (bHLH)-type transcription factor, which shares similarities with Arabidopsis BES1-interacting myc-like proteins (BIMs). BIM1, along with its counterparts BIM2 and BIM3, collaborates with BES1 in the transcriptional regulation of BR-induced genes [23]. Our results indicate that MAPK6 may also interact with BIM2, implying potential involvement in regulating BR signaling.

The presence of BR-related proteins in the interactome hints at a potential involvement of BR signaling in the *rmd* wavy root phenotype. This is further supported by the induction of wavy root growth in both rice and *Arabidopsis* by the BR analog eBL [17,18,19]. Hence, our initial focus was on confirming the IP-MS interaction data for the BR-related proteins and MAPK6. The results of yeast two-hybrid (Y2H) assays corroborated interactions between both WT MAPK6 and the MAPK6*^sor8^*mutant protein with MKK4 and BIM2 (Figure 3B). Additionally, bimolecular fluorescence complementation (BiFC) assays conducted with MAPK6 indicated that the interaction with BIM2 took place in the nucleus, aligning with BIM2’s function as a transcription factor, whereas the interaction between MAPK6 and MKK4 occurred in both the nucleus and cytoplasm (Figure 3C). Despite the lack of impact of the *sor8* mutation on the ability of the MAPK6 protein to interact with BIM2 and MKK4, Western blot analysis revealed a significant decrease in MAPK6 protein in *sor8* and *rmd sor8* mutants (Appendix A), suggesting that the mutation disrupts the protein’s stability.

The interactions between MAPK6 and BR signaling proteins prompted us to examine BR sensitivity of rice seedling roots using epibrassinolide (eBL), a synthetic BR. Upon treatment with eBL, both WT and *rmd sor8* roots exhibited a wavy phenotype, with the intensity of waviness increasing at higher eBL concentrations (Figure 4A,C). Furthermore, eBL enhanced the waviness of *rmd* roots, suggesting that an excess of BR exacerbates the root waviness phenotype (Figure 4A,C). By contrast, *sor8* roots exhibited lower sensitivity to exogenous eBL compared to WT and *rmd* roots (Figure 4A,C). Notably, treatment with a BR biosynthesis inhibitor, brassinazole (BRZ), eliminated the wavy phenotype in *rmd* (Figure 4B,D). These results suggest that the wavy root phenotype of *rmd* is influenced by BR signaling, and that MAPK6 may have a positive impact on BR signaling in root waviness.

### 2.4. BR Signaling Is Enhanced in the rmd Mutant

Elevated levels of BR signaling result in nuclear accumulation of active BZR1 (BRASSINAZOLE RESISTANT1), a key transcription factor in the BR signaling cascade, whereas reduced BR levels impede its nuclear localization [24,25,26,27]. To study the effects of *rmd* deficiency on BR response, we generated a new *rmd* knockout in a transgenic line expressing GFP-tagged BZR1 [28] (Appendix A–C). Our findings demonstrated a marked augmentation in nuclear BZR1 accumulation in the BZR1-GFP *rmd* line (Figure 5A,B), suggesting BR signaling is enhanced in the *rmd* mutant.

When the signaling output of BR is strong, the expression of positive BR signaling or biosynthetic genes, such as *DWARF*, *DWF4* (*DWARF4*), *D2* (*Ebisu dwarf*/*DWARF2*), *D11* (*DWARF11*), *CPD* (*CONSTITUTIVE PHOTOMORPHOGENIC DWARF*), *BRI1* (*BRASSINOSTEROID-INSENSITIVE1*), and *DLT* (*DWARF AND LOW-TILLERING*) [29,30,31], may be suppressed through negative feedback regulation. The expression levels of these genes were assessed in WT and *rmd* roots, revealing downregulation of them in *rmd* (Appendix A). These results further support the notion that BR signaling in *rmd* is enhanced and suggests that RMD serves as a negative regulator of BR signaling. Additionally, upregulation of *RMD* expression following exogenous eBL treatment (Figure 5C and Appendix A) suggests that *RMD* itself is subjected to feedback regulation by BR.

**Figure 5 ijms-25-10543-f005:**
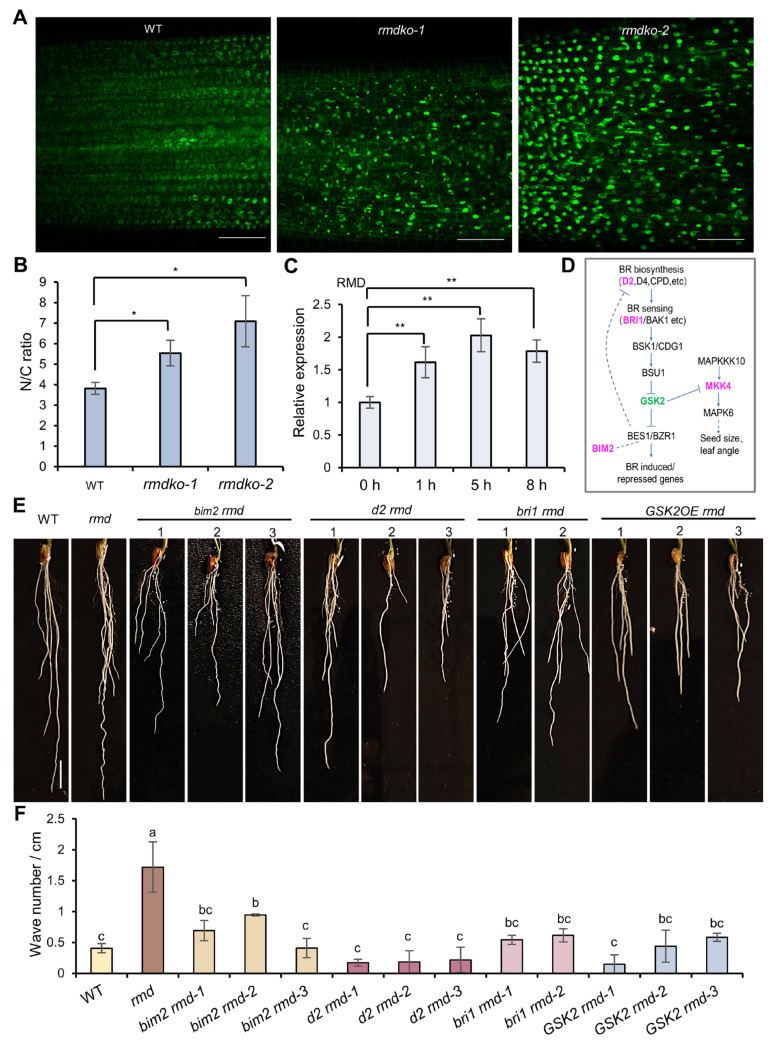
BR signaling is enhanced in *rmd* lines. (**A**) BZR1-GFP location in the root transition–elongation zone (TEZ) of 7-day-old WT and *rmd* knockout (*rmdko*) lines. Scale bars, 50 μm. (**B**) Nuclear/cytoplasm (N/C) ratios of GFP signal intensity in roots as shown in (**A**). Data are means ± s.d. of three biological replicates (*n* = 18 cells) for each genotype. Asterisks indicate significant differences relative to WT (Student’s *t*-test; * *p* < 0.05). (**C**) *RMD* expression in WT roots at different time points after treatment with 0.2 μM BL. Expression levels relative to *UBQ5*. The expression level in 0 h was set as “1”. Data are means ± s.d. (*n* = 3 replicates). Asterisks indicate significant differences (Student’s *t*-test (** *p* < 0.01). (**D**) A simplified BR signaling pathway plus MAPKK module known to function in rice seed size and leaf angle control through BR [23,32,33,34,35,36]. Colors indicate proteins chosen for subsequent analysis. (**E**) Root phenotypes of WT, *rmd,* and T_1_ lines of *bim2*, *d2*, *bri1* knockout and *GSK2* overexpression in *rmd* lines; numbers represent different lines. Scale bar, 1 cm. (**F**) Statistical data of wave number per unit root length of WT, *rmd*, and T_1_ lines of *bim2*, *d2*, *bri1* knockout and *GSK2* overexpression in *rmd* lines. Data are means ± s.d. (*n* = 3 biological replicates). Different letters represent significant differences determined by ordinary one-way ANOVA with Tukey’s multiple comparisons test, *p* < 0.05).

### 2.5. Genetic Evidence of RMD in the BR Signaling Pathway

BR signaling has been extensively researched in *Arabidopsis*, and it is proposed that rice possesses conserved BR biosynthesis and signal transduction pathways [23,32,33,34,35,36] (Figure 5D). In the *rmd* mutant, we individually mutated five genes in these pathways to suppress BR synthesis and signaling (Appendix A).

We initially knocked out *D2*, which encodes a cytochrome P450 involved in BR biosynthesis [37,38], and *BRI1*, which encodes a BR receptor involved in signal transduction [32]. Additionally, *GSK2*, which represses transcription factors BZR1 and BES1 that activate BR-responsive genes in response to high BR levels [32,36], was overexpressed. Notably, all genetic combinations (*d2 rmd*, *bri1 rmd*, and *GSK2OE rmd*) suppressed the wavy root phenotype (Figure 5E,F), supporting the association between RMD and BR signaling. In addition, BIM2 and MKK4, which interact with MAPK6 (see Figure 3), were also targeted for knockout. Consistent with the above findings, we observed substantially reduced wavy phenotypes in the double mutants (*mkk4 rmd* and *bim2 rmd*) (Figure 5E,F and Appendix A). While BIM2’s function in rice is largely unknown, we here report on a noticeable attenuation of the eBL-induced wavy root phenotype in *bim2* (Figure 4A and Appendix A). These genetic and pharmacological results demonstrate that BIM2 positively regulates BR signaling in the rice root, connecting MAPK6 with BR signaling via BIM2.

### 2.6. RMD and MAPK6 Affect Root Exploration and Avoidance of Obstacles

Circumnutation plays a crucial role in facilitating primary root colonization of soil [2,39]. To examine the impact of impaired circumnutation on root colonization in *rmd* roots, an experiment was performed to observe the ability of roots to navigate past a perforated plate positioned 1.5 cm below the seed (Figure 6A). The primary roots of WT, *rmd sor8*, and *sor8* plants were able to effectively locate and pass through the hole, whereas the primary roots of *rmd* plants exhibited coiling behavior upon encountering the plate, resulting in a significantly lower success in finding the holes (Figure 6B,C). These results suggest that the *rmd* mutation reduces the capacity of primary roots to explore obstacle surfaces and traverse them, a defect that can be rectified by *sor8* mutation. RMD and MAPK6 thus act in concert to regulate rapid root surface exploration and obstacle avoidance, thereby promoting efficient primary root penetration and seedling establishment.

## 3. Discussion

Our previous study has shown that mutations in RMD cause wavy root growth in rice [4,40], but the mechanism underlying this phenomenon and its impacts on root growth remained obscure. Here, we demonstrate that the wavy root phenotype is caused by enhanced BR signaling and is modulated by the OsMKK4-OsMAPK6-BIM2 module.

We identified a novel role for RMD in circumnutation, as evidenced by the observation that mutation of RMD still allows roots to execute circumnutation but prevents straightening of bends, resulting in a preserved circumnutation track in the final root structure (Figure 1). These observations suggest that the bending and straightening processes in root circumnutation may be controlled independently, where RMD functions in the straightening process. It is reported that the wavy phenotype of Arabidopsis roots cultured on an inclined surface transitions to a circling phenotype when the force of gravity is excluded with clinostats or in true space [41]. Similarly, in the Arabidopsis gravitropic defective mutant *rgr1*, the wavy root phenotype on a inclined agar plate was also transformed into a circling phenotype [42]. We also observed that the *rmd* root grew in circles when reaching the container bottom. We propose that before reaching the bottom of the container, the root grows downward in a helical manner in the direction of gravity (which is 1 G in the direction of growth), i.e., it repeats the process of bending–straightening. Whereas the *rmd* root, lacking the straightening process, externalizes the trajectory of the circumnutation, i.e., the wave (helical) pattern. Upon reaching the container bottom, its growth is restricted to the horizontal plane, analogous to the absence of gravity in the direction of growth, as the component of gravity is zero in this direction and therefore cannot be functional. Thus, each wave (helical) bending, which originally occurred in a three-dimensional space, is continuously displayed/compressed at the bottom plane, giving rise to the tendency to form a circle pattern. It is noteworthy that the *rmd sor8* mutant does not exhibit a substantial variation in gravitropism compared to *rmd* after a 90° gravistimulation (Appendix A), implying that gravitropic sensing and circumnutation may be independently regulated. Furthermore, some theoretical models have attributed root waving to root–gel interaction/friction [20,43,44]. Based on our analysis of *rmd*, we propose that at least part of the previously identified wavy root mutants may share a similar mechanism with *rmd*, where their phenotypes are primarily due to defects in root circumnutation. Further detailed studies of these mutants are warranted to elucidate their specific mechanisms.

Hormones such as ethylene, cytokinin, and auxin have been known to impact root circumnutation [2]. Here, we demonstrate that BR also contributes to this process. As *rmd* lines have elevated BR signaling (Figure 5), its defect in circumnutation can be rescued by either genetically or pharmacologically inhibiting BR signaling or synthesis, underscoring the importance of BR homeostasis for proper circumnutation. Loss-of-function OsMAPK6 was reported to reduce BR sensitivity [45]. Our results further confirm that the primary root of *OsMAPK6*’s mutant is less sensitive to BR than WT (Figure 4) and that OsMAPK6 directly interacts with the BR signaling component BIM2 in the root (Figure 4), thereby elucidating MAPK6’s role in BR signaling and its novel function in root circumnutation. Combining our genetic and pharmacological results (Figure 4 and Figure 5 and Appendix A), we propose that the wavy root phenotype observed in *rmd* mutants is attributable to heightened BR activity, with mutation of *MAPK6* in *sor8* mitigating this phenotype by exhibiting diminished BR sensitivity. In *Arabidopsis*, the microtubule-associated protein CLASP tethers sorting nexin 1 vesicles to microtubules, sustaining and promoting BR signaling by aiding in the retrieval of endocytosed BRI1 receptors to the plasma membrane [46]. Exogenous eBL induces alterations in cytoskeletal organization and PIN2 polar localization similar to auxin, resulting in a wavy root phenotype. On the other hand, an altered actin cytoskeleton configuration caused by the mutation in *ACTIN2* induces constitutive BR response, leading to changes in PIN2 localization and, ultimately, a wavy root phenotype, comparable to WT plants treated with eBL [20,21]. Our previous studies revealed altered PIN2 localization and auxin distribution in *rmd* mutants [40]. Our results therefore suggest a connection between BR signaling and auxin response mediated by the cytoskeleton. We propose that during root circumnutation, PIN2 and auxin can act as downstream effectors modulated by BR to establish an auxin gradient in the root that ultimately determines its growth pattern.

Notably, recent single-cell sequencing (scRNA-seq) studies of Arabidopsis root under BR treatment, conducted by the Philip N. Benfey group, have provided crucial insights into the regulatory mechanisms underlying BR-mediated elongation [47]. By using time-series scRNA-seq, the study identified the elongating root cortex as a site of brassinosteroid-mediated gene expression, and found that brassinosteroids promote a shift from proliferation to elongation, which was associated with increased expression of cell wall-related genes. Two brassinosteroid-induced downstream transcription factors, HAT7 (HOMEOBOX FROM ARABIDOPSIS THALIANA 7) and GTL1 (GT-2-LIKE 1) were further identified as key regulators controlling cell elongation along cortex trajectories by inducing the expression of cell wall-related genes. BES1 and GTL1 were also found to interact and control a common set of targets induced by brassinosteroids [47]. It is plausible that during root circumnutation, BR may also exert substantial control over the oscillatory expression of cell wall-related genes during root circumnutation, thus contributing to the rhythmic changes in cell expansion that drive this movement. The MAPK6-BIM2 module identified in our study may be involved in the targeting and regulation of downstream genes. Considering that multiple hormones such as ethylene, cytokinin, auxin, and BR found in this study are involved in root circumnutation, the interplay between BR and other hormones in root circumnutation, as well as the identification of downstream executing factors will be worthy of in-depth research in the future.

The successful penetration of roots into soil is critical for plant anchoring and soil resource absorption, particularly during the early stages of primary root development, thereby ensuring uniform field seedling establishment and crop production [2,39,48,49,50]. It has long been hypothesized that circumnutation is important in facilitating the root exploration of soil in avoiding obstacles [2,39]. This hypothesis is supported by our study on RMD and OsMAPK6 (Figure 6), and their link to BR signaling, in the context of root circumnutation. These components work together to assure efficient obstacle avoidance and root penetration, thus facilitating rapid root colonization at early seedling stage, and promoting deep root growth. We propose that our results might provide insights for agricultural practice, especially towards current trends of straw returning [51] and dry direct seeding into soil [52,53]. More specifically, early seedling vigor, especially root anchorage and expansion, are important traits in direct-seeded rice [54,55], and may be compromised by physical barriers such as undegraded straw in the soil. Our research on root obstacle avoidance provides a theoretical basis for breeding highly competitive crops suited for current agricultural production trends.

Moreover, root waving phenotypes on a slanted agar surface were altered in both hydrotropic mutants *nhr1* (*no hydrotropic response 1*) [56] and *miz1* (*mizu-kussei 1*) [57], compared to the wild type. This hints that root circumnutation may play crucial role in hydrotropism and other tropisms, such as trophotropism. Considering that the molecular mechanisms underlying root circumnutation remain largely unexplored, future research focusing on the role of root circumnutation in various root tropisms and environmental adaptation will be highly interesting and potentially provide novel insights into plant root development.

## 4. Materials and Methods

### 4.1. Plant Materials and Growth Conditions

Wild type (WT) rice for these experiments was *Oryza sativa* L. ssp *japonica* cultivar 9522. The existing *rmd* mutant is in 9522 background, with a CAAGG to T replacement in the 11th exon of *RMD* [4]. The *rmd sor8* line was isolated from ethyl methanesulfonate (EMS)-induced mutations of *rmd*, while the single *sor8* mutant was selected from the segregated progeny of a cross between *rmd sor8* and WT rice. In the BC6F2 generation of *rmd sor8* × *rmd*, separately mixed samples of the *rmd*-type and *rmd sor8*-type populations from the segregating population were collected and used for genome resequencing, along with the original *rmd sor8* line, and for candidate position analyses.

Unless otherwise stated, rice seeds were surface-sterilized prior to germination with 1.5% (*v*/*v*) H_2_O_2_ for 6 h, 30% (*v*/*v*) bleach for 25 min, then washed 3 times with sterilized water. Seeds were germinated in sterilized water for 3 d at 28 °C in the dark, then transferred to a transparent growth box with the appropriate medium: tap water; soil; or ½ Murashige and Skoog (1/2 MS) medium pH 5.8 solidified with 0.6% (*w*/*v*) agar (MS agar). Rice grown in liquid medium were restrained in transparent 96-well plates with the bottom removed. Seedlings were grown at 28 °C for 5–7 d in 16 h light/8 h dark.

### 4.2. Plasmid Construction and Plant Transformation

For genetic *sor8* complementation, the 8.78 kb genomic *pMAPK6*:*MAPK6* DNA fragment (2.8 kb promoter upstream of the start codon and entire open reading frame minus the stop codon) was amplified from WT gDNA and cloned into a plant binary vector pCAMBIA1301-35S-eGFP, so as to replace the 35S promoter.

For *OsGSK2* overexpression in *rmd*, the *OsGSK2* coding sequence (CDS) was amplified from WT cDNA and cloned into pTCK303 under control of the maize *ubiquitin* promoter.

For knocking out *BIM2*, *MKK4*, *D2*, *BRI1* and *RMD*, gene-specific 20 bp guide RNAs (gRNAs) were designed with CRISPR-P 2.0 (http://crispr.hzau.edu.cn/CRISPR2/) [58] combined with manual BLAST searches and screening in NCBI. If the first nucleotide of the gRNA was not an adenine, one was artificially added to ensure transcription from the rice U3 snoRNA promoter (U3p). These gRNAs were cloned into CRISPR/Cas9 vector pRGEB32 [59] at the *Bsa*I site using the Golden Gate Assembly method [60] with slight modifications. For each gene target, 4 nt overhangs (5′-GGCA and 5′-AAAC) were added to the 5′ end of sense and antisense gRNAs, respectively. The gRNAs were then diluted to 100 pmol μL^−1^, mixed in a 1:1 ratio, annealed at a temperature gradient (95 °C 30 s, 72 °C 2 min, 37 °C 2 min, 25 °C 2 min), and finally ligated into *Bsa*I-digested pRGEB32 by T4 DNA ligase.

All constructs were verified by DNA sequencing, and introduced into WT, *rmd*, *rmd sor8*, or BRZ1-GFP marker line rice calli by *Agrobacterium tumifaciens*-mediated transformation [61]. Transgenic plants were identified by PCR using specific primers for corresponding genes or the antibiotic resistance markers. Primers used are listed in Appendix A.

### 4.3. Propidium Iodide (PI) Staining

The 2 mm sections of root tips or the mature zone (2 cm from the root tip) of 7 d seedlings were cut and stained with 1 μM PI in 1 × phosphate-buffered saline (PBS) for 10 min, then washed in PBS 3 times for 10 min, and subjected to Leica TCS SP5 confocal laser scanning microscopy using 488 nm excitation and 610 ± 20 nm emission filters.

### 4.4. Gravity Response

Seedlings were grown at 28 °C in tap water in 16 h light/8 h dark for 3 d. Seedlings were placed on 1% agar and allowed to grow vertically for 3 h, then placed horizontally for 6 h and photographed. The tip angle of the primary root was defined as the angle formed between the direction of the root tip and the horizontal base line. Photos were analyzed with ImageJ software (version 1.54d).

### 4.5. Hormonal Treatments

Seedlings were grown in tap water. 2,4-Epibrassinolide (eBL; Sigma, Steinheim, Germany) was added to a final concentration of either 0.01 μM, 0.2 μM or 0.02 μM, with reference to Zhang et al. 2009 and Chen et al. 2015, and adjusted according to preliminary testing with the cultivar used in this study [62,63]. Brassinazole (BRZ; Selleck, Houston, TX, USA) was added to a final concentration of either 5 μM or 20 μM, with reference to Qiao et al., 2017 and the test with the cultivar used in this study [30]. As eBL and BRZ were diluted from ethanol storage stocks, the same amount of ethanol was added to all treatments, including the mock.

### 4.6. Immunoprecipitation (IP) and Mass Spectrometry (MS)

WT and *pMAPK6*:*MAPK6-GFP* complemented *sor8* lines were grown at 28 °C in tap water for 7 d. About 2 g of roots was harvested and ground in liquid nitrogen. Proteins were extracted with 4 mL extraction buffer (20 mM Tris–HCl, pH 7.5, 150 mM NaCl, 1 mM EDTA (ethylenediaminetetraacetic acid) (Sigma, Steinheim, Germany), 0.5% Triton X-100 (BBI Life Sciences, China), 1 mM PMSF (phenylmethylsulfonyl fluoride) (BBI Life Sciences, Shanghai, China), and 1 × cOmplete^TM^ EDTA-free Protease Inhibitor Cocktail (Roche, Basel, Switzerland)). The mixture was sonicated 3 times for 15 s on ice, with 15 s pauses. After 30 min incubation on ice, the homogenized sample was centrifuged 3 times at 20,000× *g* at 4 °C. After transferring the supernatant to a fresh tube, 20 μL washed anti-GFP beads (supplier) was added, incubated at 4 °C for 2 h, and the beads were washed 3 times again with washing buffer (as extraction buffer). Proteins were eluted with lithium dodecyl sulfate (LDS) buffer (GenScript, Piscataway, NJ, USA), separated by SurePAGE (GenScript, Piscataway, NJ, USA), and visualized with Coomassie Brilliant Blue staining. Bands were excised and sent for MS analysis (Shanghai BioTree, Shanghai, China).

### 4.7. Yeast Two-Hybrid (Y2H) Assays

Full-length *MAPK6* and *MAPK6^sor8^* CDS were fused in-frame with the sequence encoding the GAL4 DNA-binding domain of the bait vector pGBKT7, while full-length *OsBIM2* and *OsMKK4* CDS were cloned into the prey vector pGADT-7. Bait/prey pairs were co-transformed into *Saccharomyces cerevisiae* strain AH109, as previously described [64]. Transformants were selected on SD plates lacking Trp and Leu, and yeast colonies were patched in duplicate onto SD/-Trp-Leu-His-Ade/X-α-gal, SD/-Trp-Leu-His/X-α-gal, and SD/-Trp-Leu/X-α-gal, to observe interactions. Primers for vector construction are described in Appendix A.

### 4.8. Bimolecular Fluorescence Complementation (BiFC) Assays

Full-length *MAPK6* CDS was cloned into pXY106-YN, while full-length *OsBIM2* and *OsMKK4* CDS were cloned into pXY104-YC. Five-week-old *Nicotiana benthamiana* leaves were co-infiltrated with *A. tumefaciens* GV3101 strains carrying the desired constructs. After 2 d incubation in the dark, fluorescence signals were observed using a Leica TCS SP5 confocal laser scanning microscope (Leica Microsystems, Wetzlar, Germany) using 488 nm excitation and 525 ± 50 nm emission filters. Primers used for vector construction are described in Appendix A.

### 4.9. Subcellular Localization

The BZR1-GFP marker line was kindly provided by Prof. Jiuyou Tang. Images were taken with a Leica TCS SP5 confocal laser scanning microscope, at 488 nm excitation and 490–530 nm emission. To measure the nuclear:cytoplasmic (N/C) ratio of GFP signal, average fluorescence intensity was determined over a fixed area (about half a nucleus in size) in both the nucleus and cytoplasm, analyzed by ImageJ software (referred to in Ibanez et al. [24]). For each genotype, 6 cells from each of 3 points along the root (upper, middle and lower) from 3 different plants were analyzed.

### 4.10. Gene Expression Analysis

For hormonal analysis, seedlings were grown in tap water for 7 d, at which point eBL was added to a final concentration of 0.2 μM. With reference to Tang et al. and the experimental repeatability of our treatment trials, we collected root samples at 0 h, 1 h, 5 h and 8 h post treatment [28].

Total RNA was extracted from roots in triplicate using the TRIZOL reagent (Invitrogen, Carlsbad, CA, USA), according to the manufacturer’s instructions. cDNA was synthesized using FastKing RT Kit (with gDNase; TIANGEN, Beijing, China) from 1 ug total RNA. qRT-PCR analysis was performed on a CFX96 (Bio-Rad, Hercules, CA, USA) machine using SYBR Green Premix Pro Taq HS qPCR Kit II (Accurate Biology, Hunan, China). *UBQ5* mRNA was used as an internal control. Primers used for qRT-PCR are described in Appendix A.

### 4.11. Perforated-Plate Assays

Seedlings were grown at 28 °C in MS agar for 5 d, in a growth box containing a dark plastic plate, with 2.5 mm holes (diameter) spaced 13 mm apart, set 1.5 cm beneath the top plane of the gel (Figure 6A). At least 9 individuals of each genotype were sown in each of 3 technical replicates, and individuals that exhibited abnormal growth of the primary root were excluded from subsequent analysis. The percentage of seedlings with primary roots passing through or around the perforated plate were counted.

### 4.12. Observation of Root Growth Dynamics

Seedlings were grown at 28 °C in MS agar for 3 d in a transparent acrylic. Roots were then photographed every 10 min for 3 h by a fixed camera with fixed position. The dynamic video was generated by cutting all the photos of each species to a fixed size and stacking the photos on the video processing software Capcut (v12.8.0).

### 4.13. Multiple Sequence Alignment

The protein sequences of MAPK6 were obtained from uniport, NCBI or Phytozome. Multiple sequence alignment was conducted using ClustalW within MEGA7 (version 7.0.21) with the default parameters, and the result was beautified and presented using Genedoc 2.7 software (version 2.7).

### 4.14. MAPK6 Protein Detection

The 7 d roots were ground to fine powder in liquid nitrogen, mixed with 2 × SDS loading buffer (250 mM pH 6.8 Tris-HCl, 10% SDS (*w*/*v*), 50% glycerol (*v*/*v*), 5% β-mercaptoethanol (*w*/*v*), 0.5% (*w*/*v*) bromophenol blue) in equal volumes, boiled at 95 °C for 10 min and centrifuged at 12,000 rpm for 5 min, and then the supernatant was taken for protein separation on a 10% SDS–PAGE gel (EpiZyme, PG212, Beijing, China). Immunoblotting was performed using anti-MAPK6 antibody (Beijing Protein Innovation, AbP80140-A-SE, Beijing, China) and the anti-α-tubulin antibody (Beyotime, AT819, Shanghai, China) was used as loading control.

### 4.15. Statistical Analysis

Two-sided unpaired Student’s *t*-tests were used to calculate significance of the difference between two groups in Microsoft Excel 2016. One-way ANOVA test with Tukey’s post-test analysis was used when comparing multiple groups (GraphPad Prism version 9.0 software). All data are presented as mean ± s.d.

### 4.16. Accession Numbers

Sequence data can be found in GenBank under the following accession numbers: *MAPK6/SOR8*, LOC_Os06g06090; *GSK2*, LOC_Os05g11730; *BRI1*, LOC_Os01g52050; *D2*, LOC_Os01g10040; *DWARF*, LOC_Os03g40540; *DWF4*, LOC_Os03g12660; *D11*, LOC_Os04g39430; *CPD*, LOC_Os10g08580; *BAK1*, LOC_Os08g07760; *ILI1*, LOC_Os04g54900; *UBQ5*, LOC_Os01g22490; *BZR1*, LOC_Os07g39220; or in the MSU database under the following accession numbers: *RMD*, Os07g0596300; *BIM2*, Os02g0726700; *MKK4*, Os02g0787300; *DLT*, Os06g0127800.

Accession numbers for MAPK protein sequences used in multiple sequence alignment are the following: Os (*Oryza sativa*) Q84UI5, 1At (*Arabidopsis thaliana*) Q39026, Sc (*Saccharomyces cerevisiae*) P41808, Hs (*Homo sapiens*) Q16659, Rn (*Rattus norvegicus*) P27704, Mm (*Mus musculus*) Q61532, Br (*Barchydanio rerio*) Q0H1F2, Pa (*Pongo abelii*) Q5R7U1, Ch (*Capra hircus*) A0A452G716, Oc (*Oryctolagus cuniculus*) G1T2T1 in UniProt; Ta (*Triticum aestivum*) XP_044449614.1, Zm (*Zea mays*) NP_001152745.2, Gm (*Glycine max*) XP_003532933.1, Hv (*Hordeum vulgare*) XP_044971508.1, Sb (*Sorghum bicolor*) XP_002467591.1, Si (*Setaria italica*) XP_004983829.1, Bd (*Brachypodium distachyon*) NP_001266884.1, Pp (*Physcomitrium patens*) XP_024390775.1, 2At (*Amborella trichopoda*) XP_011623655.1, Dm (*Drosophila mojavensis*) XP_002004089.2, Ce (*Caenorhabditis elegans*) NP_001366708.1, Ss (*Sus scrofa*) XP_001925326.1, Bt (*Bos taurus*) XP_002691011.2 in NCBI; Pt (*Populus trichocarpa*) Potri.007G139800.1.p in Phytozome.

## Figures and Tables

**Figure 1 ijms-25-10543-f001:**
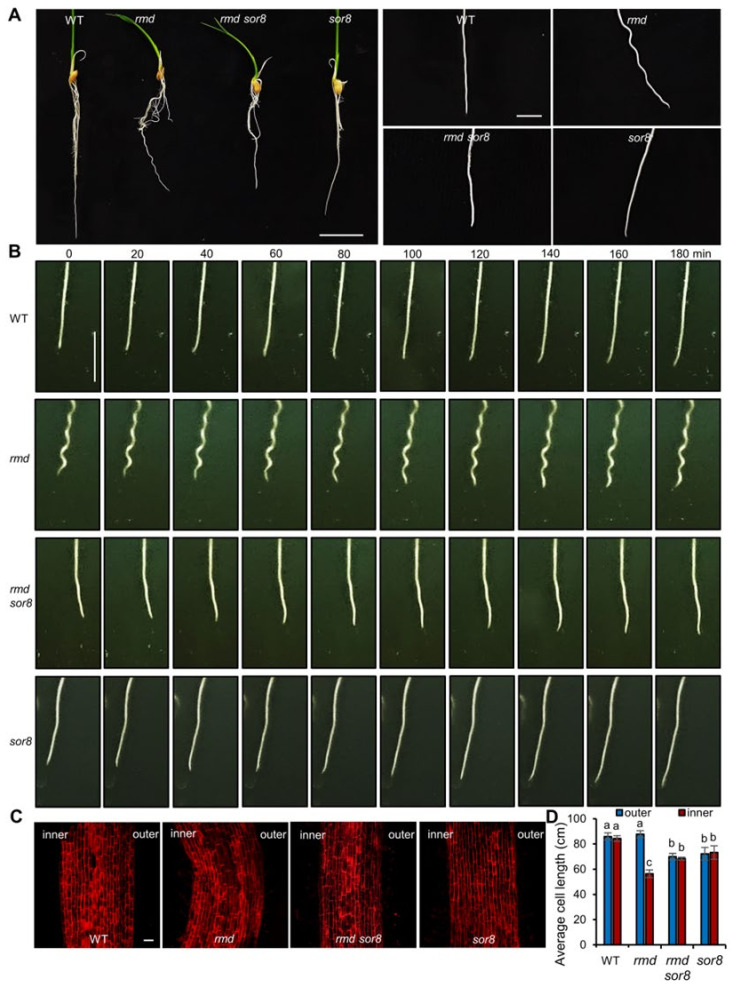
RMD and its suppressor SOR8 participate in the control of root circumnutation. (**A**) Root phenotypes of 7-day-old wild type (WT), *rmd*, *rmd sor8*, and *sor8* lines grown in water. Each root tip is magnified on the right. Scale bars, 2 cm (original image); 0.5 cm (magnified image). (**B**) Movement during primary root circumnutation in four different genotypes at 20 min intervals over 3 h growth (3-day-old plants). Scale bar, 1 cm. (**C**) Cell morphology of four lines visualized by propidium iodide staining. Scale bar, 50 μm. (**D**) Cell lengths of root epidermis cells on the outer and inner sides of the bent-root region. Data are means ± s.d. of three biological replicates (*n* = 21 cells) for each genotype. Different letters represent significant differences determined by ordinary one-way ANOVA with Tukey’s multiple comparisons test, *p* < 0.05.

**Figure 2 ijms-25-10543-f002:**
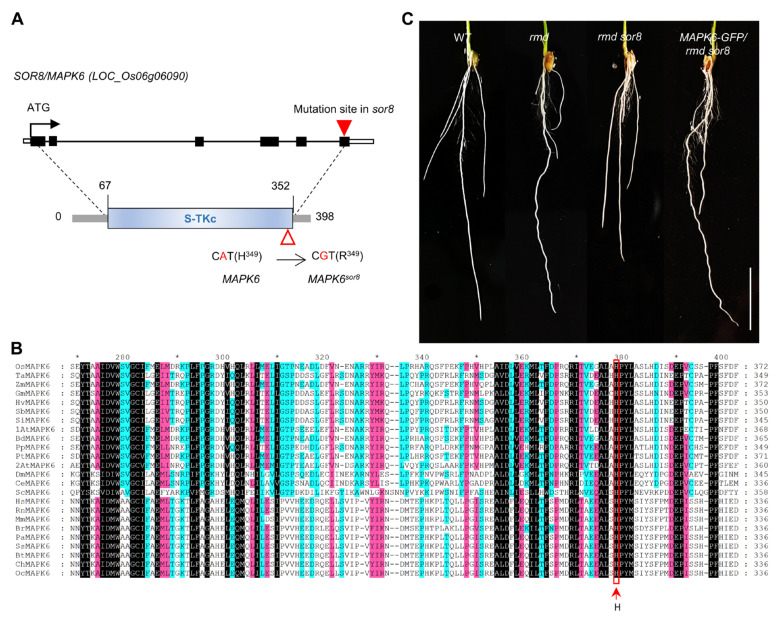
Cloning and genetic verification of *SOR8*. (**A**) Schematic diagram of *SOR8* structure and *sor8* mutation. The black rectangles depict *SOR8* exons; the black line depicts introns; and hollow boxes at each end depict untranslated regions. The solid red triangle shows the mutation site in *sor8*, creating a 1 bp substitution (A to G). Below, the blue rectangle represents the functional serine/threonine kinase catalytic (S-TKc) domain of MAPK6. The hollow red triangle shows the position of the mutation, an amino acid replacement from His^349^ to Arg^349^. (**B**) Amino acid alignment of regions of MAPK6s corresponding to *sor8* mutation between different species generated by MEGA7 followed by Genedoc 2.7. Os (*Oryza sativa*), Ta (*Triticum aestivum*), Zm (*Zea mays*), Gm (*Glycine max*), Hv (*Hordeum vulgare*), Sb (*Sorghum bicolor*), Si (*Setaria italica*), 1At (*Arabidopsis thaliana*), Bd (*Brachypodium distachyon*), Pp (*Physcomitrium patens*), Pt (*Populus trichocarpa*), 2At (*Amborella trichopoda*), Dm (*Drosophila mojavensis*), Ce (*Caenorhabditis elegans*), Sc (*Saccharomyces cerevisiae*), Hs (*Homo sapiens*), Rn (*Rattus norvegicus*), Mm (*Mus musculus*), Br (*Barchydanio rerio*), Pa (*Pongo abelii*), Ss (*Sus scrofa*), Bt (*Bos taurus*), Ch (*Capra hircus*), Oc (*Oryctolagus cuniculus*). Black-, pink-, and blue-shade highlights represent 100%, 80%, and 60% conserved amino acids between all species, respectively. The red arrow indicates the conserved “H” (Histone) site corresponding to the *sor8* mutation. (**C**) Root phenotypes of wild type, *rmd*, *rmd sor8*, and the complemented *pMAPK6:MAPK6-GFP*/*rmd sor8* line. Scale bar, 2 cm.

**Figure 3 ijms-25-10543-f003:**
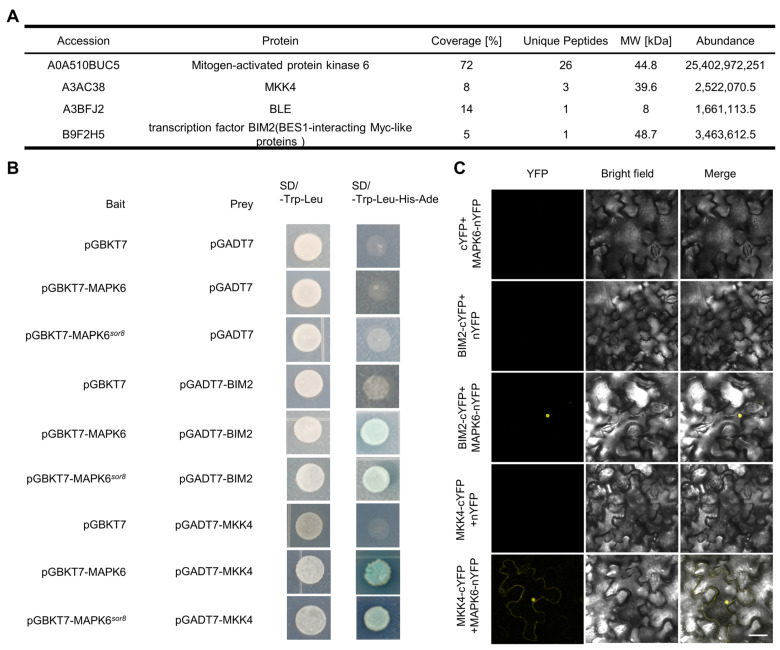
Detection and verification of MAPK6-interacting proteins. (**A**) Mass spectrometry analysis showing MKK4, BLE1, and BIM2 were co-purified with MAPK6 in vivo. (**B**) Yeast two-hybrid assays indicate MAPK6 and MAPK6*^sor8^* interact with BIM2 and MKK4. The yeast clones were cultured on SD medium lacking Leu and Trp, and interaction detected on SD medium lacking Leu, Trp, His, and Ade and containing X-α-gal. (**C**) Bimolecular fluorescence complementation assays showing interactions between MAPK6–BIM2 and MAPK6–MKK4 in tobacco leaves. Scale bar, 50 μm.

**Figure 4 ijms-25-10543-f004:**
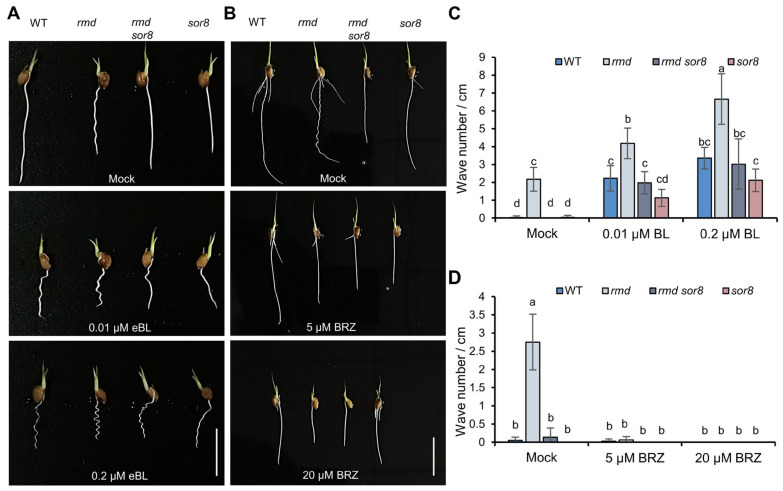
Treatment with brassinosteroid (BR) analog eBL and biosynthesis inhibitor BRZ. (**A**,**B**) The root phenotypes of four lines after 5 days’ growth in water with or without (mock) eBL (**A**) or BRZ (**B**). Scale bars, 2 cm. (**C**,**D**) Number of waves per unit root length in plants grown as in (**A**,**B**). Data are means ± s.d. (*n* = 6 replicates). Different letters represent significant differences determined by ordinary one-way ANOVA with Tukey’s multiple comparisons test, *p* < 0.05.

**Figure 6 ijms-25-10543-f006:**
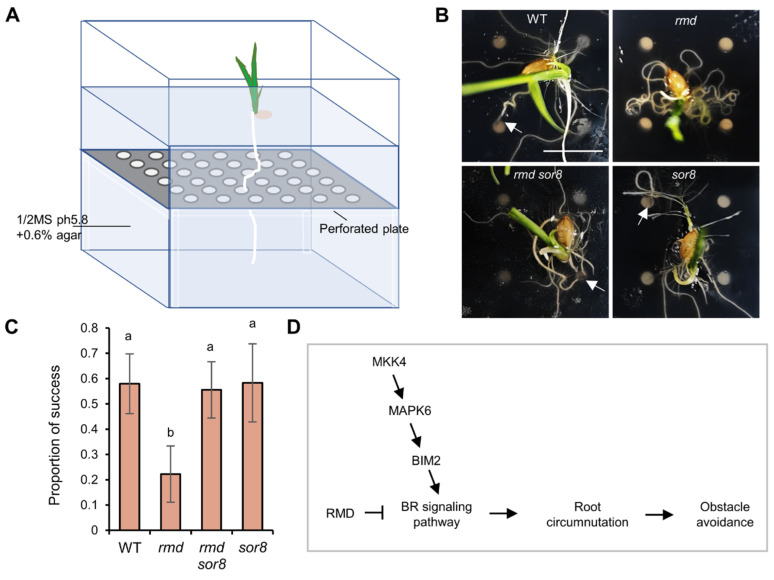
RMD and MAPK6 modulate root exploration of obstacle surfaces. (**A**) Schematic diagram of the experimental system for perforated plate assays. (**B**) Top-down image of representative roots in four lines after 5 days’ growth in the perforated plate assay. Arrows indicate primary roots. Scale bars, 1 cm. (**C**) Proportion of seedlings (*n* > 8) of each genotype having roots which penetrate the perforated plate. Data are means ± s.d. of three technical replicates, each with *n* > 8 individuals. Different letters represent significant differences determined by ordinary one-way ANOVA with Tukey’s multiple comparisons test, *p* < 0.05. (**D**) Diagram of proposed BR signaling pathway regulating root circumnutation and obstacle avoidance. Arrows denote participation in the process they direct. Blocked line indicates suppression of the process it point.

## Data Availability

All data are contained within the main text or Appendix A.

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
