# Peer review of "RMD and Its Suppressor MAPK6 Control Root Circumnutation and Obstacle Avoidance via BR Signaling"

_ijms, 2024, doi:10.3390/ijms251910543_

Round 1

Reviewer 1 Report

Comments and Suggestions for Authors

In this manuscript RMD and Its Suppressor MAPK6 Control Root Circumnutation and Obstacle Avoidance via BR Signaling” by Dong et al., this article found that rice actin binding protein RMD contributes to root rotation, and this rotation defect depends on brassinosteroid (BR) signaling. The author further inhibited BR signaling through pharmacological (BR inhibitor) or genetic (knockout of BR biosynthesis or signaling components) manipulation, which can rescue root defects in RMD. At the same time, it was found that mutations in MAPK6 inhibited BR signaling and restored normal root rotation in RMD. This study is of great significance for promoting root development and root biology research.

Moreover, I have some other concerns, listed as following:

1.         In the Introduction, it is necessary to provide a concise and clear explanation of the purpose of this study or the key issues that this study aims to address, as well as how we plan to proceed, rather than what problems we have already solved. Relevant content should be added in the last paragraph.

2.         In the structure of the paper, there are both results and discussion sections. Therefore, when writing the results section, it is sufficient to clarify the main results of this study, without citing other papers or references.

3.         Please pay attention to the writing format of gene names. Generally, gene names need to be italicized, and the entire text needs to be consistent, without the phenomenon of italicization and overall alternation.

4.         Please pay attention to the writing format of the unit. Eg, change “ul” to “uL”. Line 395, “…(ethylenediaminetetraacetic acid)…”, inconsistent font format.

5.         All references should be formatted according to the requirements of the journal. Currently, many papers have incomplete citations, please check and revise them one by one.

Comments on the Quality of English Language

Suggest finding a professional to polish the language, as many sentences are too colloquial.

Reviewer 2 Report

Comments and Suggestions for Authors

The aim of the paper is to assess the role of rice actin-binding protein RMD in root circumnutation. The Authors demonstrated that root circumnutation defects in rmd depend on brassinosteroid (BR) signaling, which is elevated in mutant roots. Suppressing BR signaling via pharmacological (BR inhibitor) or genetic (knockout of BR biosynthetic or signaling components) manipulation rescues root defects in rmd. The study evidenced that RMD and MAPK6 control root circumnutation by modulating BR signaling to facilitate early root growth.

In my opinion, the manuscript is very well organized and prepared. The results are interesting for plants biologists. It should be emphasized both the high scientific level as well as the illustrative/graphic layer of the manuscript. The only drawback is insufficiently described sections of Materials and Methods, as well as the Discussion could be rewritten to consider the new citations in the field of research topic and in-depth interpretation of the results. In addition, I formulated the following improvements in the revision stage of the manuscript processing:

-           The Authors should explain in the manuscript on what basis the concentrations of 2,4-Epibrassinolide and brassinazole during hormone treatments were selected.

-            Regarding the subchapter “4.10. Gene Expression Analysis”, it should be explained why time-points at 1 h, 5 h and 8 h were chosen. Moreover, in the supplementary file should be included the figures presenting the results of Melting Curve Analysis (post-PCR checking out if there are no non-specific amplicons, e.g. primer-dimer adducts). Furthermore, there is no information how RNA samples were assessed before gene expression studies (maybe some electrophoretic analyses? Spectrophotometric measurements?).

-          Minor editing of English language is required.

Comments on the Quality of English Language

Minor editing of English language is required.
